# Advances of Single-Cell Protein Analysis

**DOI:** 10.3390/cells9051271

**Published:** 2020-05-20

**Authors:** Lixing Liu, Deyong Chen, Junbo Wang, Jian Chen

**Affiliations:** 1State Key Laboratory of Transducer Technology, Aerospace Information Research Institute, Chinese Academy of Sciences, Beijing 100190, China; liulixing16@mails.ucas.ac.cn (L.L.); dychen@mail.ie.ac.cn (D.C.); 2School of Electronic, Electrical and Communication Engineering, University of Chinese Academy of Sciences, Beijing 100049, China; 3School of Future Technologies, University of Chinese Academy of Sciences, Beijing 100049, China

**Keywords:** single-cell analysis, protein characterization, conventional approaches, microfluidic technologies

## Abstract

Proteins play a significant role in the key activities of cells. Single-cell protein analysis provides crucial insights in studying cellular heterogeneities. However, the low abundance and enormous complexity of the proteome posit challenges in analyzing protein expressions at the single-cell level. This review summarizes recent advances of various approaches to single-cell protein analysis. We begin by discussing conventional characterization approaches, including fluorescence flow cytometry, mass cytometry, enzyme-linked immunospot assay, and capillary electrophoresis. We then detail the landmark advances of microfluidic approaches for analyzing single-cell protein expressions, including microfluidic fluorescent flow cytometry, droplet-based microfluidics, microwell-based assay (microengraving), microchamber-based assay (barcoding microchips), and single-cell Western blotting, among which the advantages and limitations are compared. Looking forward, we discuss future research opportunities and challenges for multiplexity, analyte, throughput, and sensitivity of the microfluidic approaches, which we believe will prompt the research of single-cell proteins such as the molecular mechanism of cell biology, as well as the clinical applications for tumor treatment and drug development.

## 1. Introduction

As the physical basis for all life and the main component of living organisms, proteins dominate or participate in almost all biological activities and biological functions like providing structural supports, molecule transportations, cell growth and adhesion, signal transductions, catalytic biochemical processes, etc. [1,2]. Under the controls of internal genes and external environments, the differences in protein expressions affect cell differentiations, nerve conductions, immune responses, and disease occurrence, which is a crucial indicator of changes in life activities [3,4]. Therefore, protein expression analysis is critical for the studies of cellular molecular mechanisms, clinical diagnosis and treatments, and drug developments [5]. In the past few decades, various methods have been developed for protein analysis, such as gel electrophoresis [6], immunoassay [7], chromatography and mass spectrometry [8], and Raman imaging [9]. These methods provide a comprehensive understanding of the biological functions of different proteins, which facilitate the developments of molecular biology and medicine [10]. However, most of these conventional approaches are limited to protein analysis at tissue levels and only able to measure population-averaged protein expressions from large amounts of cells [11], masking the single-cell heterogeneity within a population [12,13]. As a result, many rare but critical individual cells are typically overlooked in conventional studies though these cells play essential roles in, for example, cancer metastasis and stem cell differentiation. Although single-cell genomic and transcriptomic analysis with high throughputs have developed rapidly to address the issue of cellular heterogeneity in recent years, studies have located poor correlations between RNA and protein levels in single cells [14]. Due to the stochasticity of gene expressions, variations occur in RNA and protein copy numbers of cells with the identical gene, which indicates the disconnection between single-cell proteomic and transcriptomic analysis and the necessity of single-cell proteomic analysis. Single-cell protein analysis enables protein analysis at the single-cell level and provides a feasible approach to distinguish and identify those rare but important single cells from large average populations, facilitating the corresponding studies related to fundamental mechanisms, disease developments, and drug therapies [15].

The big challenges of single-cell protein analysis are the enormous complexity and low abundance of the proteome [16,17]. Thus, single-cell protein analysis must be high-multiplexity, high-throughput, and high-sensitivity to provide quantitative information [18,19]. Some of the conventional technologies can solve the problems by single-cell separation and signal analysis (such as fluorescence or mass spectrometry) for protein detection. Besides, microfluidics provides a reliable technology for manipulating cells at very tiny volumes, thus can effectively fit single-cell analysis.

In this review, we mainly summarize the recent two-decade advances of various single-cell protein analysis approaches and techniques. We first present the developments of several key conventional approaches including fluorescence flow cytometry, mass spectrometry flow cytometry, enzyme-linked immunospot assay, and capillary electrophoresis. Then we focus on the latest advances enabled by microfluidic technologies for single-cell protein detection, including microfluidic fluorescent flow cytometry, droplet-based microfluidics, microwell-based assay (microengraving), microchamber-based assay (barcoding microchips), and single-cell Western blotting. We discuss the performance of each system in terms of multiplexity, analyte (e.g., membrane, intracellular, and secreted proteins), throughput and sensitivity, comparing advantages and limitations, and providing our perspectives on the potential development directions of future studies.

## 2. Conventional Approaches

### 2.1. Fluorescence Flow Cytometry

Fluorescence flow cytometry is the golden-standard approach for profiling of proteins at the single-cell level, which enables measurements of fluorescence characteristics of single cells or any other particles in a fluid stream when they pass through a light source [20,21,22]. Specifically, when single cells stained with fluorescent labelled antibodies rapidly travel through the detection region in the flow chamber, stained cells are excited by a laser, and a detector measures the emitted fluorescent intensities [23,24]. By building calibration curves using beads that have been coated with proteins under precise controls, fluorescent intensities could be translated to single-cell protein expressions [25,26] (Figure 1A).

Since its emergence in the 1960s [30], as the most established method for single-cell protein analysis, fluorescence flow cytometry made remarkable technological advancements and was featured with high throughputs and multiplexing [31]. Based on the working principle of continuous flow, it enables high-throughput detection of measuring ~10^4^ cells per second [32]. With fluorescent labelled antibodies, it is capable of analyzing ~20 multiplexing protein parameters for membrane and intracellular proteins associated with signaling pathways in single cells [33].

Furthermore, fluorescence flow cytometry has transformed from a primitive cell counter to a powerful tool for semi-quantitative analysis, especially for analyzing pathways underlying diseases [34], discovering surface markers [35], and processing drug screening [36]. For example, it is a generally accepted method to determine the type of leukemia by detecting CD series differentiation antigens on the surface of cell membranes and estimating the proportions of immune cell subtypes [37]. More specifically, Chattopadhyay et al. found diversely complex phenotypic patterns in total CD8+ T cells with a modified flow cytometry of 17 fluorescence emissions based on fluorescent quantum dots [38].

However, due to the rapid flowing of samples, neither measurement of secreted proteins nor the dynamic monitoring of cells over time is easy to achieve. The multiplexing capacity is limited due to spectral overlap even if fluorescence compensation is conducted. Due to the significant loss during the sample preparation process, mass populations of single cells are required, making it difficult to detect rare samples. In addition, because the cells are exposed to physical stressors such as fluidic pressure and laser beams, this can damage the cellular integrity and hamper recovery [39].

### 2.2. Mass Cytometry

Mass cytometry is a technique that integrates flow cytometry and mass spectrometry to analyze single-cell protein expressions with distinct transition element isotopes labelled antibodies on and within cells rather than fluorescence [8,27]. Stained single cells are pushed into a nebulizer, ionized through an argon plasma, and separated by the ions mass-to-charge ratio. Based on the time-of-flight mass spectrometer, results for each cell’s constituent ions are sampled, transformed, and integrated to electric signals, which can be further quantified as single-cell protein measurements [27,40] (Figure 1B).

Compared to fluorescence flow cytometry, mass cytometry uses heavy metal element labels to avoid cross-talks among channels in fluorescence and reduces background noise interferences, which enables high multiplexed detections of surface and intracellular proteins with over 40 different proteins simultaneously measured [41]. Nolan’s group used mass cytometry to profile primary human bone marrow cells with multiple parameters simultaneously for phenotype analysis. They monitored signaling behaviors of cell subpopulations based on subtype-specific surface markers [27]. As for the throughput, it depends on the time-of-flight sampling resolution, mass cytometry enables the measurement up to ~10^3^ cells per second, inferior to that of fluorescent labelled analysis approaches. A new method named mass-tag cellular barcoding was developed by Nolan’s group, which improved throughput by using n metal ion tags to multiplex up to 2n samples and applied to characterize signaling proteins and pathways in human peripheral blood mononuclear cells [40]. Besides, compared to quantum-efficient fluorophores, mass reporters show lower sensitivities, which makes it difficult to measure low expressed proteins in single cells. Moreover, since mass cytometry requires ionization, cellular recovery and preserving integrity are still infeasible. The common limitations for both mass cytometry and fluorescence flow cytometry are incapable of analyzing secreted proteins at the single-cell level, for lack of approaches that maintain small molecules and binding agents associated with the cells.

In order to obtain information on cell localization and interactions, several improved methods based on mass cytometry came into being later [42]. Imaging mass cytometry is applied to tissue analysis with a high-resolution laser ablation system to time-of-flight mass cytometry, which achieves measurements of over 100 markers possible with the availability of additional isotopes [43]. Compared with mass cytometry, which is only applied to cell suspensions, imaging mass cytometry allows spatial information of cells through tissue analysis. Another approach, multiplexed ion beam imaging, is a secondary ion imaging method that operates an ion beam to release metal ion reporters and uses mass spectrometry to quantify, which can simultaneously determine more than 100 targets [44]. As the advances and complements to mass cytometry, these methods can achieve higher resolutions and multiplexing parameters for single-cell protein analysis.

### 2.3. Enzyme-Linked Immunospot Assay

Enzyme-linked immunospot assay, developed in the 1980s, is a quantitative approach for detecting secreted protein at the single-cell level [45,46]. Single cells are localized on a plate coated with capture antibodies against specific secreted proteins. After stimulation to cells, the secreted proteins are captured by the primary antibodies and the signal is further amplified by secondary antibodies. Each visual spot represents a single cell expressing the target proteins and intensities of spot indicate secretion levels of target proteins [28,47] (Figure 1C).

Enzyme-linked immunospot is highly sensitive for detection of secreted proteins with a six spots per 10^5^ cells detection limit [48]. It is widely used in the studies of immune responses, such as detecting cytokine-secreting cells [49,50] and monitoring immune system activations [51,52]. Herr et al. proposed a fast enzyme-linked immunospot assay to quantitate CD8 + T lymphocytes of HIV patients and proved a reliable detection of T cell reactivity due to previous exposure to HIV [53]. Karlsson et al. made a comparison of enzyme-linked immunospot and flow cytometry to assay CMV and HIV-1 proteins in chronically HIV-1-infected patients. Though results of T cell responses were statistically correlated between two approaches, it showed consistently lower results in the enzyme-linked immunospot assay, which suggested that it was preferable to detect low-level responses [54]. Kornum et al. presented an enzyme-linked immunospot assay to test hypocretin in CD4+ T-cells and indicated that epitope frequency was lower than the detection limit (1:10,000 cells) among peripheral CD4+ T-cells from narcolepsy type I patients [55]. However, this approach can only detect no more than three secreted proteins simultaneously. Compared with flow cytometry, the throughput is insufficient because it is a static assay.

### 2.4. Capillary Electrophoresis

Capillary electrophoresis is a separation and detection approach based on a high-voltage electric field in a micron capillary whose inner diameter is compatible with single cells [29,56]. Single cells are injected into the capillary under electromigration or pressure and then lysed via chemical, optical, or electrical methods resulting in lysed ions of diverse levels of migration properties. Combined with electrochemical analysis, laser-induced fluorescence, mass spectrometry, and other technologies, according to the migration times, the detector outputs an electrophoretic spectrum, in which a peak corresponds to a type of protein. The abundance of each protein can be reflected to the statistics of the peaks, such as height or area [57,58,59] (Figure 1D).

Capillary electrophoresis exhibits a high sensitivity and only requires ultra-low injection. Schultz et al. described a capillary electrophoresis with laser-induced fluorescence, realizing a detection limit of 3 nM or ~6 fg injection for secreted insulin, which demonstrated the capability of rapidly determining a low level of protein in single cells [60]. Sobhani et al. presented an ultrasensitive fluorescence detection that proteins were separated and analyzed by 2-dimensional capillary electrophoresis. They used the tool to characterize the single-cell proteins and biogenic amines from the murine macrophage cell line, revealing large variations in component expressions among single cells [61]. As a technology well-suited for analysis of small heterogeneous samples, capillary electrophoresis was reported by Phillips et al. to measure protein tyrosine phosphatases in single cells of human epidermoid carcinoma, which provided a powerful tool for the analysis of human biopsy specimens [62]. In spite of these advantages, several intermediate steps, such as cell injection, lysis, and separation, result in the whole process being time-consuming and having low throughput.

## 3. Microfluidic Approaches

Microfluidics is a technology to process and manipulate small amounts of fluids (10^-9^–10^-18^ L) based on microfabricated channels [63]. Due to the dimensional compatibility with biological cells, microfluidic systems capable of miniaturization, integration, and parallelization have become an ideal platform for the analysis of single-cell proteins [64,65]. In the recent two decades, some microfluidic approaches have been developed and made great improvements on single-cell analysis of protein expressions.

### 3.1. Microfluidic Flow Cytometry

Microfluidic flow cytometry is a miniaturized version of flow cytometry for analysis of a small number of cells and enables integration of sample handling and single-cell analysis on a single microfluidic chip, where protein analysis is conducted [66]. By integrating microfluidic fabrication, optical sources and fluorescence detection together, microfluidic flow cytometry facilitates single- cell protein analysis and achieves quantification based on calibration curves (Figure 2).

Quake’s group developed a microfabricated flow cytometer for sorting various biological cells in 1999 [67], and since then, microfluidic flow cytometry for single-cell protein analysis has developed rapidly. Preckel et al. demonstrated a commercially available microfluidic system for analysis of protein expressions of fluorescently stained primary cells, with a small number down to 625 cells per sample [68]. In order to achieve dynamic detections, a microfluidic platform combining multi-color flow cytometry and fluorescence microscopy was proposed by Wu et al. for probing signaling events spanning multiple timescales and intercellular locations [69]. Chen et al. reported an improved microflow cytometry platform based on a constriction channel enabling the quantification of numbers of multiple intracellular proteins simultaneously from tens of thousands single cells from both tumor cell lines and patient samples [70,71].

**Figure 2 cells-09-01271-f002:**
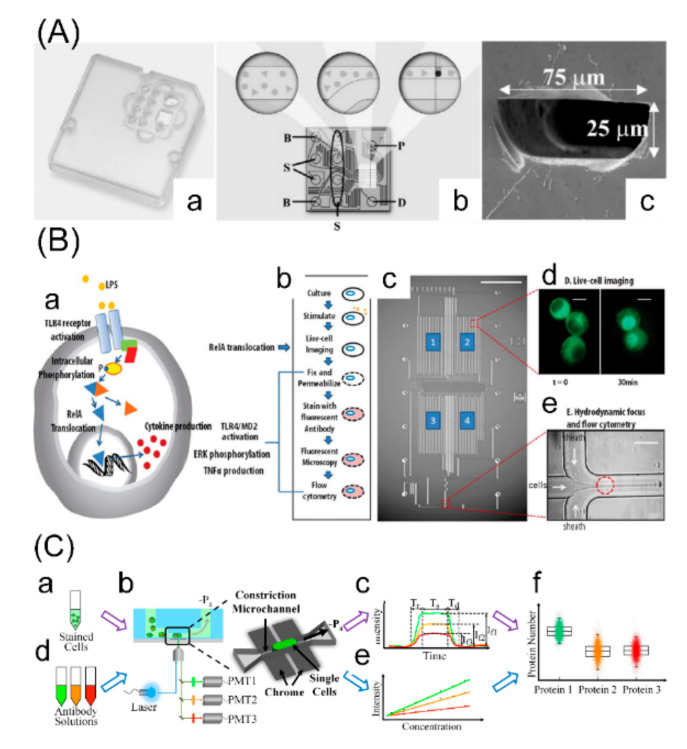
Microfluidic flow cytometry for single-cell protein analysis. (**A**) A commercially available microfluidic flow cytometry for analysis of protein expression with a small number down to 625 cells per sample. (a) Schematic of the microfluidic chip; (b) layout of the microfluidic glass chip with sample wells (S), buffer wells (B), the well for the reference dye (D), and the priming well (P); (c) cross-section micrograph of a channel with dimensions of 25 × 75 μm after bonding top and glass plate. Adapted with permission from [68]. (**B**) A microfluidic chip for global profiling of cellular pathways. (a) TLR4 signaling events occur at different timescales and subcellular locations; (b) shows the workflow procedures integrated and performed on the chip shown in (c); all the representative events in the cell diagram can be profiled using both fluorescent microscopy (d) and flow cytometry (e). Adapted with permission from [69]. (**C**) An improved microflow cytometry platform based on a constriction channel for absolute quantification of multiple intracellular proteins. Cells stained with multiple fluorescent labelled antibodies (a) are aspirated into the constriction microchannel with excited fluorescent signals detected by photomultiplier tubes (b); for each travelling cell, time coordinated fluorescent pulses are obtained with fluorescent levels (c); the calibration curves are obtained by the gradient solutions of multiple types of fluorescent labelled antibodies (d,e); based on raw parameters and calibration curves, numbers of multiple types of intracellular proteins are obtained (f). Adapted with permission from [71].

Microfluidic imaging flow cytometry is a modified method to collect spatial information at a high throughput. McKenna et al. presented a parallel microfluidic cytometer with 384 parallel flow channels for protein localization in a yeast model with a high throughput of several thousand events per second [72]. Furthermore, Holzner et al. proposed a microfluidic imaging flow cytometer for the ultra-high-throughput (60,000 and 400,000 cells per second for blur-free fluorescence and brightfield detection, respectively) quantitative imaging analysis of cytoplasmic proteins in human cells. It was capable of multi-parametric fluorescence quantification and subcellular localization analysis of cellular structures down to 0.5 μm with microscopy image quality [73].

Compared to conventional flow cytometry, microfluidic flow cytometry greatly reduces the amount requirements of samples which is helpful for applications in studying rare samples such as primary cells and rare tumor cells. In addition, it can obtain intracellular spatial information of single cells with a high throughput and is featured with the capacity of absolute quantification. The microfluidic flow cytometry improves some features; however, it has several similar limitations as conventional flow cytometry, i.e., the limited multiplexing capacity and incapability of quantifying secreted proteins.

### 3.2. Droplet-Based Microfluidics

Droplet-based microfluidics allows the quantification of secreted proteins, thereby overcoming the major limitations for protein analysis by microfluidic flow cytometry [74,75]. Typically, single cells and reagents, including fluorescent probes and target antibodies, are encapsulated simultaneously in the pico- or nanoliter water-in-oil emulsion-droplets. After incubation, fluorescent labelled antibodies bind to the secretions within the droplets. Subsequently, the droplets are loaded into a continuous flow channel, and the signal intensities are quantified, enabling a high-throughput droplet generation and protein analysis [76] (Figure 3).

By confining single cells within tiny rooms by droplets, droplet microfluidics has worked as a well-established tool in single-cell protein analysis. Huebner et al. described an approach based on picoliter microdroplets initially, performing high-throughput screening by detecting the enzyme alkaline phosphatase expressed by *Escherichia coli* cells [77,80]. Weitz’s team presented droplet-based microfluidics for high-throughput analysis of proteins released from or secreted by cells, screening individual enzyme expressions at a rate of ~10^7^ per hour [81,82]. To realize the absolute quantification of tiny protein concentrations, a new approach that combines a proximity ligation assay and droplet-based digital PCR for protein quantification was developed by Albayrak et al. They counted both endogenously (CD147) and exogenously (GFP-p65) expressed proteins from hundreds of single cells [78]. Stoeckius et al. introduced a method of cellular indexing of transcriptomes and epitopes by sequencing (CITE-seq) based on droplet-based microfluidics to analyze protein and RNA expressions simultaneously for thousands of single cells. They exploited this method to detect multiplexed protein markers of cord blood mononuclear cells and enabled classifications of immune subpopulations [83]. Furthermore, Dhar et al. described a droplet-based microfluidic system integrated with vortex capture for estimating single-cell protease activities, which concentrated rare circulating tumor cells >10^6^-fold from whole blood into 2-nL droplets and characterized the collagenase enzymes with a high-sensitivity of ~7 molecules per droplet [79].

As a popular approach of single-cell protein analysis, droplet-based microfluidics is capable of compartmentalizing highly controllable activities for a high-sensitivity analysis of intracellular, membrane, and especially secreted proteins. Nevertheless, it is a low efficient detection approach for limited cell encapsulation by the Poisson distribution, which would cause invalid analysis of empty or multiple cells in a droplet. Besides, changes in the microenvironments of single cells in droplets may cause unclear effects on cell activities in comparison to in vivo situations.

### 3.3. Microwell-Based Assay (Microengraving)

The microwell-based assay (microengraving) is a technique to monitor the temporal dynamics of secreted proteins from single cells based on microwells (~1 nL) in a large array [84]. In this method, single cells are distributed in large-array wells with antibody-coated microengraved substrates, and the corresponding antibodies capture the secreted proteins. After short periods of incubation, the slide with captured proteins is removed and analyzed by the conventional enzyme-linked immunosorbent assay [85] (Figure 4).

After Love’s group first proposed this technology in 2006, a series of microengraving approaches have been applied in single-cell protein analysis. To improve the sensitivity, a hybridization chain reaction was integrated into this platform to amplify signals resulting from sandwich immunoassay for simultaneous detections of three secreted proteins, improving the sensitivity by an average of 200-fold compared to direct fluorescence detections [86]. Furthermore, it can provide a dynamical scope when immune responses of white blood cells (such as T-cells and B-cells) are monitored [87,89,90,91]. For example, Jia et al. presented a study of evaluating multiple parameters based on microengraving to analyze the protein-conjugate vaccine responses in adult nonhuman primates of B-cells. Compared to the enzyme-linked immunospot assay, the nanowell-based assay increases the sensitivity with a 10^6^-fold higher concentration of analytes from given cells and enables the recovery of cells for further genetic analysis [91]. To detect low numbers of proteins with a broad dynamic range, another microwell-based assay design named “single molecule array” was presented by Walt et al. They demonstrated a wide range of expression of prostate-specific antigens with variation over several orders of magnitudes, revealing that genetic instabilities in cancer cells can affect protein expressions [88].

In all, the microengraving method is a powerful dynamics tool for single-cell protein analysis with advantages of high sensitivity, wide dynamic range, and capability of cell recovery. However, it characterizes only secreted proteins, but not membrane and intracellular proteins. Additionally, due to the spectral overlaps of colorimetric fluorescent proteins, its multiplexing capacity is limited to no more than four proteins. In addition, the throughput is also a limitation, because of the limited size of the microchip and the filling rate of single cells in each well requiring complex manipulations.

### 3.4. Microchamber-Based Assay (Barcoding Microchips)

In the same period, other than microwell-based assay, microchamber-based assays (barcoding microchips) function as an effective approach for analyzing proteins in single cells [92]. As an approach of absolute quantification in the number of protein molecules, this approach utilizes control microvalves to isolate single cells within known volumes of microchambers that contain capture antibodies in a barcode array. When proteins are captured, each microchamber containing an entire barcode can be quantitatively analyzed via a surface-bound immune sandwich assay (Figure 5).

Heath’s team first demonstrated this method and a series of follow-up studies. Ma et al. presented a single-cell barcode chip for quantitative measurements of over 10 secreted proteins from single cells and applied the chip to quantify the effector molecules of T cells, observing the functional heterogeneity in cytotoxic T lymphocytes [96]. Apart from secreted proteins, Shi et al. described a new barcode chip for quantification of cytoplasmic and membrane proteins, and the microchip evaluated protein interactions related to PI3K signaling pathway mediated by EGF receptor [97]. Moreover, Wang et al. extended the function to the detection of comprehensive analytes (including membrane, intracellular, and secreted proteins) based on a modified barcode chip [93]. To further increase the multiplexity, Lu et al. designed a combination of spatial spectrum coding and microchambers, and realized detection of 42 secreted proteins. Through a comparative analysis of differentiated macrophages between different stimulations, distinct functional heterogeneity was exposed [94]. Additionally, another barcoding microchip was used to examine secreted proteins in isolated cell pairs to identify the most stable separation distances between two cells [95].

This approach has been conducted with advantages of precise quantification, comprehensive analyte detection and multiplexing capacity, and a commercial instrument of “Isoplexis” has been developed. Despite these advantages, it also has some limitations. Due to the complex fabrication of microvalves on the chip, the effective area of the barcoding microchip is restricted, resulting in a limited detection throughput, as well as the requirement of sophisticated operations. Additionally, a balance is needed that either maintains the multiplexing capacity or detection sensitivity; that is to say multiplexing capacity would decrease assay sensitivity.

### 3.5. Single-Cell Western Blotting

Existing methods are almost antibody-based assays, which may cause a false-positive signal because of the non-specific binding from antibody cross-reactivity. As a recently proposed technology, single-cell Western blotting is a combination of microfluidics and conventional Western blotting to achieve protein expression analysis at a single-cell resolution [98]. Due to separation by electrophoresis before the antibody probing, it overcomes the issue of cross reactions. In single-cell Western blotting, a layer of polyacrylamide gel is coated on a glass and patterned with a large-array microwells. Single cells are dropped on the thousands of microwells and lysed in situ, and then proteins are separated by gel electrophoresis, immobilized via photoinitiated blotting, and detected by fluorescent labelled antibodies [99,100] (Figure 6).

As a young approach of single-cell protein analysis, single-cell Western blotting has developed rapidly in recent years since Herr’s group first reported it. Kang et al. described a useful protocol to measure single-cell variation in protein expressions based on single-cell Western blotting, enabling detection of more than 10 proteins in each cell during 4 h [102]. Due to cell loss, thousands of cells are required in single-cell Western blot. To solve the problem, Sinkala et al. introduced a single-cell resolution microfluidic Western blotting for multiple membrane and intracellular proteins expressions in circulating tumor cells with only two starting cells to monitor the response to therapy [101]. To improve identification specificity in single-cell Western blotting, Kim et al. established a molecular mass standard with a “solid phase” protein marker. The magnetic field was used to guide the protein-coated particles into most (>75%) microwells, accomplishing His protein marker release subsequently and protein solubilization and cell lysis simultaneously [103]. To improve analytical sensitivity and throughput, Gumuscu et al. recently introduced a hybrid single-cell Western blotting integrated with separation-encoded microparticles. The dehydrated microparticles were reduced dimensionally based on the hydrogel molding and release method, thereby enhancing the sensitivity obviously. Meanwhile, ERα expression from breast tumor cells were quantified with a reduced immunoprobing time of ~36 h based on mass transport in microparticles [104].

Although single-cell Western blotting represents a new technology for single-cell protein expression analysis, some limitations are obvious. It is a relative quantification approach due to lack of calibration and it is unable to quantify the secreted proteins. Furthermore, single-cell Western blotting has limited detection sensitivity because proteins are easily lost during processing procedures such as cell lysing, protein immobilization, and repeated antibody stripping.

## 4. Conclusions and Outlook

In this review, we summarized the key advances of conventional and microfluidic technologies for single-cell protein analysis in the past two decades, and made an approach comparison for multiplexity, analyte, throughput, and sensitivity (Table 1). The rapid developments and enormous progress of single-cell protein research offer unprecedented opportunities in studying multiplexed, high-throughput, and high-sensitivity of single-cell proteins (including membrane, intracellular, and secreted proteins). Apart from improving our understanding of the cellular molecular mechanisms (cellular heterogeneity), it is helpful for applications of clinical diagnosis, tumor treatments, and drug developments.

In the field of single-cell protein expression analysis, conventional approaches often have certain advantages, for instance, fluorescence flow cytometry—high throughput; mass cytometry—multiplexed capacity; enzyme-linked immunospot assay—high sensitivity; capillary electrophoresis—comprehensive analytes. Compared with conventional technologies, microfluidic approaches usually integrate several strengths, which makes assays of rare cells possible.

Despite the recent technological advances, the limitations of current single-cell protein analysis technologies are also obvious. From the aspect of multiplexity, current multiplexing is still not enough for whole proteomics detections (>10,000 proteins in a single cell). As for the analyte, comprehensive detections of the membrane, intracellular, and secreted proteins are a mainstream trend that most approaches are capable of only one or two specific types, while in this review only droplet-based microfluidics and barcoding microchips could simultaneously achieve detections but limited in other aspects. Throughput is another important evaluation parameter because of the analysis requiring large numbers of cells and a large amount of data; flow cytometry-based techniques usually beat other techniques in terms of throughput. In addition, high sensitivity is necessary for accurate and quantitative analysis for single-cell proteins; however, most current approaches still cannot reach the single-cell level limit of detecting single-molecule protein quantification.

In addition, in order to achieve comprehensive analysis of single-cell proteins, single-cell proteomic analysis can be combined with multi-omics (e.g., genomics, transcriptomics, or metabolomics). Increasing evidence shows that integrating multiple genetic data was essential to obtain accurate understanding of biological information [105,106]. Moreover, as an important supplementary information in addition to protein abundance, spatial information is also necessary for single-cell proteomic characterization. It includes both protein characteristics such as protein locations and cell characteristics such as cellular phenotypes and cellular dynamics. Combining information from comprehensive multi-omics and spatial-omics, a complete new insight of cellular status and heterogeneity can be obtained.

In the future work, researchers will still focus on improving multiplexity, analyte, throughput, and sensitivity uniformly based on combination, parallelization, and automation. The combination of multiple technologies can leverage the advantages of different approaches, for example, applying continuous cell flow detections in large-array microchips to increase multiplexity and throughput [107], combining droplets with signal amplification technologies to increase sensitivity, such as immunoassay [108], proximity ligation/extension assay [109,110], and sequence-topology assembly for multiplexed profiling [111]. Besides, parallelization of microchannels for single-cell processing enables increased throughput. Automation is also critical to provide commercial services of transforming technologies into a reliable and effective instrument that can apply to clinical diagnosis and treatments.

## Figures and Tables

**Figure 1 cells-09-01271-f001:**
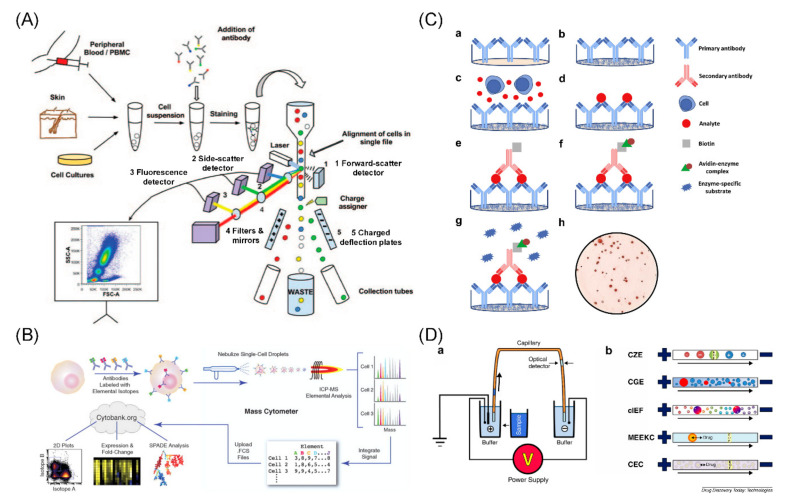
Schematics of conventional approaches for single-cell protein analysis. (**A**) Fluorescence flow cytometry. Single cells stained with fluorescent labelled antibodies rapidly travel through the flow chamber, stained cells are excited by a laser, and the emitted fluorescent intensities are measured by the detector. Additionally, the fluorescent intensities could reflect the single-cell proteins expression. Adapted with permission from [22]. (**B**) Mass cytometry. Stained single cells with element isotopes labelled antibodies are pushed into a nebulizer and ionized, and an elemental mass spectrum is acquired for each cell. The integrated elemental reporter signals for each cell can then be analyzed by flow cytometry. Adapted with permission from [27]. (**C**) Enzyme-linked immunospot. Single cells are localized on a plate coated with capture antibodies against specific secreted proteins. When the cells secrete proteins after stimulation, the secreted proteins are captured by the primary antibody and the signal is further amplified by secondary antibody. Each visual spot signal readout represents a single cell expressing the target protein and intensity of spot indicates proteins secretion level. Adapted with permission from [28]. (**D**) Capillary electrophoresis. Single cells are injected into the capillary under electromigration or pressure and then lysed via chemical, optical, or electrical methods resulting in lysed ions of diverse levels of migration properties. Combined with electrochemical, laser induced fluorescence, mass spectrometry, and other technologies, the detector outputs an electrophoretic spectrum which can reflect the protein expression. Adapted with permission from [29].

**Figure 3 cells-09-01271-f003:**
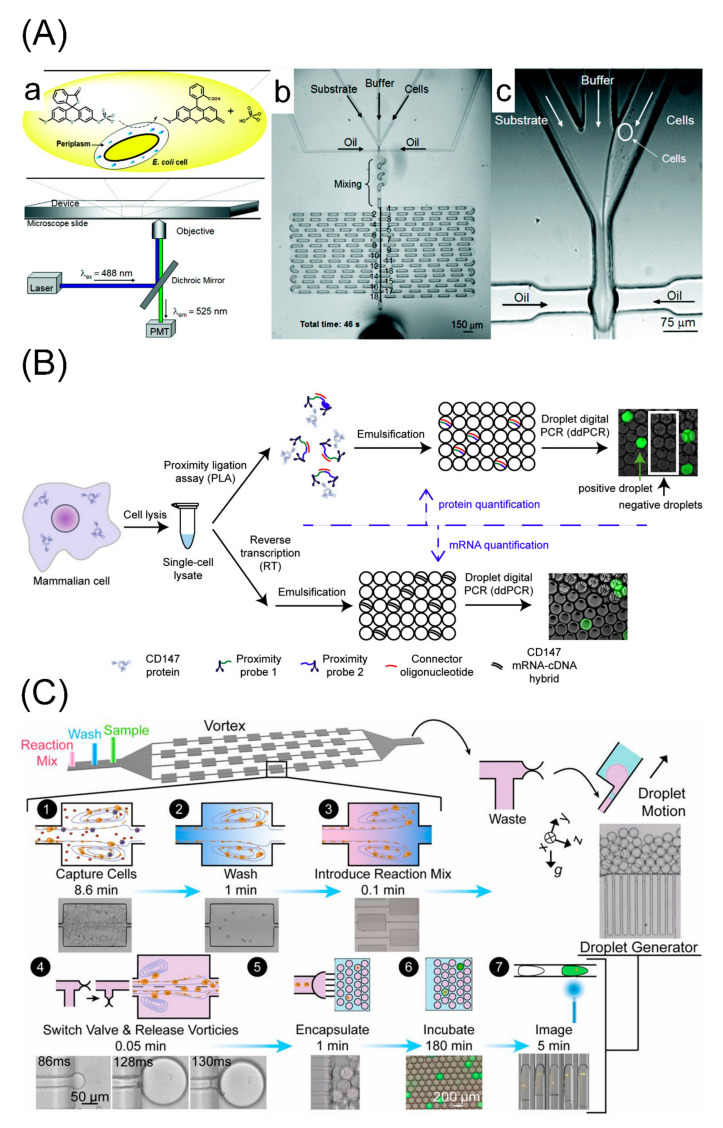
Droplet-based microfluidics for single-cell protein analysis. (**A**) A microfluidic device of picoliter droplets for enzymatic reaction. (a) Single *E**scherichia coli* and substrate 3-O-methylfluorescein-phosphates are encapsulated within single droplets where the substrate is enzymatically hydrolyzed by the target enzyme alkaline phosphatase expressed by *E. coli*, generating a fluorescent signal; (b) and (c) show the droplet formation that occurred by confluence of three aqueous inlet streams (substrate, buffer and cells). Adapted with permission from [77]. (**B**) A new approach for absolute quantification of proteins combining proximity ligation assay and droplet digital PCR. Targeted proteins are isolated, lysed, and converted to dsDNA by standard proximity ligation assay. The dsDNA is distributed among 20,000 droplets at limiting dilution. Single dsDNA molecules in the droplets are then amplified by PCR and counted by measuring the fluorescence using droplet reader based on calibration curve. Adapted with permission from [78]. (**C**) A droplet-based microfluidic system for enzyme secretion from circulating tumor cells (CTCs) based on size purification. The system isolates CTCs by size, exchanges fluid around CTCs to remove contaminants, introduces a matrix metalloprotease substrate, and encapsulates CTCs into microdroplets. The cells can then be incubated and imaged by an imaging cytometer in the droplet generator. Adapted with permission from [79].

**Figure 4 cells-09-01271-f004:**
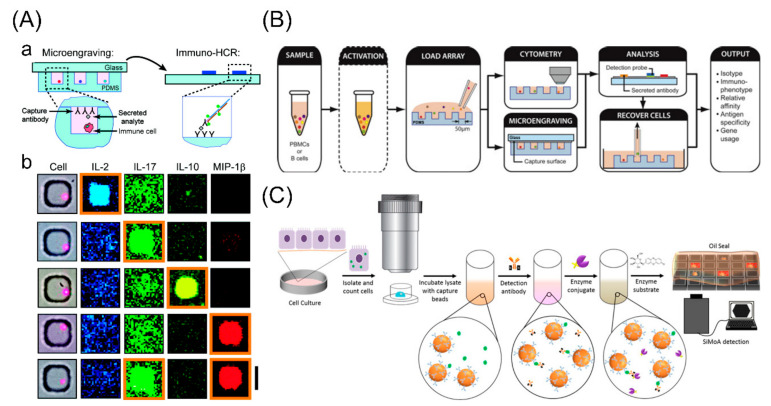
Microwell-based assay (microengraving) for single-cell protein analysis. (**A**) An integrated platform for microengraving and hybridization chain reaction. (a) Schematic illustration for detection of secreted products from single cells. Single cells are deposited onto an array of microwells on a glass slide with antibody coated. After incubation, the slide is removed, and immune-hybridization chain reaction is used to amplify the signal related to each capture event; (b) fluorescent micrographs for secreted proteins following microengraving and immune-hybridization chain reaction. Adapted with permission from [86]. (**B**) Process schematic for the integrated analysis of B cells using microengraving and on-chip cytometry. Microwells loaded with stained cell are imaged on a microscope cytometry to record the expressed phenotypes of every cell and the occupancy of each well. Microengraving can then be performed to capture secreted anti-bodies. Cells of interest can be recovered with an automated micromanipulator, and then sequenced further. Adapted with permission from [87]. (**C**) A single molecule array approach for quantifying phenotypic responses. Cultured cells are isolated, lysed, and loaded into the analyzer of single molecule array, and then incubated with capture beads, target antibody, and enzyme conjugate. The enzyme substrate is added, and the oil seal is used after the immune complex is formed on the beads, and then the imaging is detected. Adapted with permission from [88].

**Figure 5 cells-09-01271-f005:**
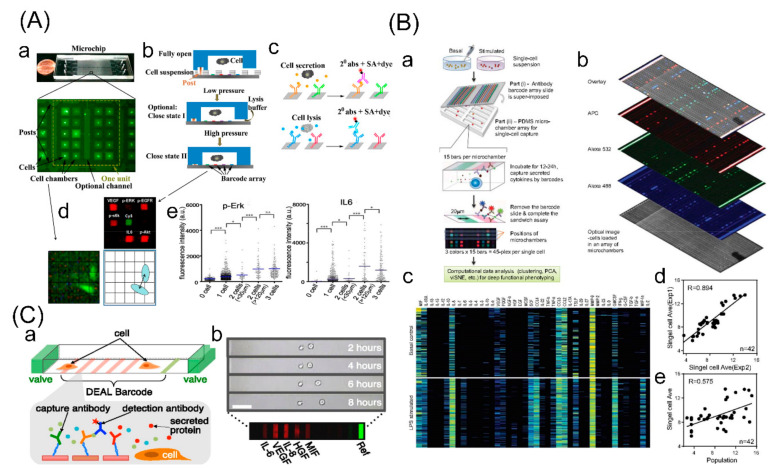
Microchamber-based assay (barcoding microchips) for single-cell protein analysis. (**A**) A single-cell barcode chip for quantitative measurements of membrane, intracellular, and secreted proteins from single cells. (a) Image of the microchip and a fluorescence micrograph of a cellular assay unit (20 microchambers); (b) workflow of the on-chip operation. Fully open: cells are loaded into the microchambers. Close-I state: microchambers are sealed by a low pressure on the microchip but lysis buffer can be introduced to the channel. Close-II state: cells are isolated completely in the microchambers from the channel by a high pressure; (c) workflow of detecting of membrane, intracellular, and secreted protein via the sandwich-type fluorescence immunoassay; (d) single-cell proteomic result of fluorescence intensity, cell numbers and cell positions; (e) fluorescence data for secreted and intracellular protein assays. Adapted with permission from [93]. (**B**) A microchamber-based platform combined with spatial and spectral encoding. (a) Workflow illustration of high-throughput profiling of single cells in basal and stimulated conditions for 42 secreted effector proteins; (b) representative optical image showing a block of microchambers loaded with U937-derived macrophage cells and the corresponding scanned fluorescence images showing protein detections with three colors; (c) representative heat maps showing single-cell protein profiles measured on U937-derived macrophages; (d) correlation of protein secretion expressions between two replicate microchip experiments at single-cell levels, and (e) between single-cell levels measured using microchips and population levels measured using conventional methods. Adapted with permission from [94]. (**C**) A barcoding microchip for identifying most stable separation distance between two cells. (a) Schematic of a single microchamber with valves and barcodes (top) and the fluorescent sandwich immunoassay protein detection scheme (bottom); (b) a representative time-lapse image of a two-cell chamber over 8 h and a typical fluorescence image of a barcode for the five assayed proteins. Adapted with permission from [95].

**Figure 6 cells-09-01271-f006:**
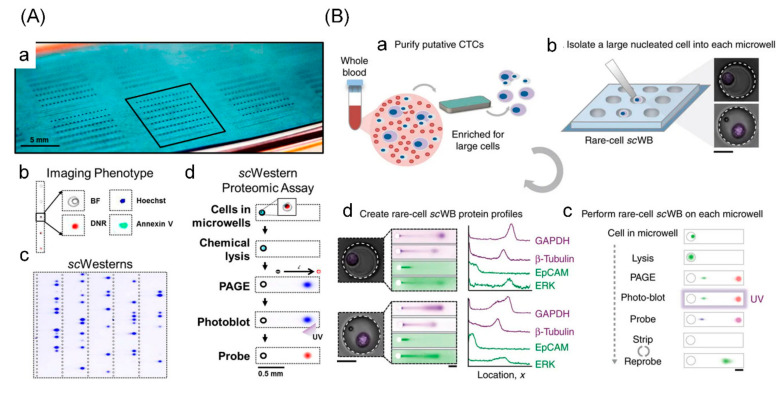
Single-cell Western blotting for single-cell protein analysis. (**A**) Schematic of single-cell phenotype imaging and Western blotting. (a) The array consists of thousands of microwells patterned in a thin layer (30 μm) photoactive polyacrylamide gel seated on a glass slide; (b) fluorescent imaging of single cells in microwells provides phenotype information; (c) single cells are lysed in situ after imaging and the lysate is used for Western blot analysis; (d) workflow of single-cell Western blotting for proteomic assay. Adapted with permission from [100]. (**B**) Single-cell Western blotting of rare cells. (a) Enrich circulating tumor cells (CTCs) from whole blood samples based on cell size; (b) deposit enriched cells into the microwell and identify each CTC by nuclear staining; (c) for each cell in microwell, proceed as in-microwell chemical CTCs lysis, single-CTC protein polyacrylamide gel electrophoresis, covalent immobilization of proteins to the gel (photo-blotting) and in-gel immunoprobing; (d) single-CTC lysate is analyzed and rounds of immunoprobing support the multiplexing of 12 proteins, thus creating a protein expression profile for each rare cell. Adapted with permission from [101].

**Table 1 cells-09-01271-t001:** Approach comparison of single-cell protein analysis for multiplexity, analyte, throughput, and sensitivity.

Approach		Multiplexity	Analyte	Throughput	Sensitivity	Reference
Conventional	Fluorescence Flow Cytometry	~20	Membrane Intracellular	~10^4^ cells/s	500/cell	[24,33,34,35,36,37,38,39]
	Mass Cytometry	~40	Membrane Intracellular	~10^3^ cells/s	N/A	[27,40,41,42,43,44]
	Enzyme-Linked Immunospot Assay	1–3	Secreted	~10^6^ cells/run	6 in 10^5^ cells	[28,47,48,49,50,51,52,53,54,55]
	Capillary Electrophoresis	1	Membrane Intracellular Secreted	~10 cells /h	3 nM	[29,58,59,60,61,62]
Microfluidic	Microfluidic Flow Cytometry	~10	Membrane Intracellular	10^4^–10^5^ cells/s	<10/cell	[67,68,69,70,71,72,73]
	Droplet-Based Microfluidics	3–4	Membrane Intracellular Secreted	10^3^–10^4^ cells/s	<10/cell	[76,77,78,79,80,81,82,83]
	Microwell-Based Assay (Microengraving)	4	Secreted	~10^4^ cells/chip	~10^3^/cell	[84,85,86,87,88,89,90,91]
	Microchamber-Based Assay (Barcoding Microchips)	42	Membrane Intracellular Secreted	~10^4^ cells/chip	~10^2^/cell	[92,93,94,95,96,97]
	Single-Cell Western Blotting	12	Membrane Intracellular	10^3^–10^4^ cells/chip	~10^4^/cell	[98,99,100,101,102,103,104]

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
