# Peer review of "Advances of Single-Cell Protein Analysis"

_cells, 2020, doi:10.3390/cells9051271_

Round 1

Reviewer 1 Report

This review describes many analytical techniques for protein expression in a single cell, covering from conventional techniques such as flow cytometry to newly developed microfluidic-based approaches such as microfluidic flow cytometry. Each technique is well explained and thus this review would be helpful for people who want to know recent advances in single cell protein analysis.

My only concern is the word "proteomic" in the title. In general, proteomics denotes another type of experiments in which a large number of unspecified proteins are analyzed at one time. In contrast, experiments shown in this review treat relatively small number of specified proteins. In addition, there is a field of proteomics in which researchers are trying to identify and study a large number of proteins from a single cell using LC-MS/MS. Thus I think it would be better to call the experiments in the review as "single cell protein analysis" rather than "single cell proteomic analysis. 

Minor point,
Line 115 : mass spectroscopy -> mass spectrometry

Author Response

We deeply appreciate your comments and suggestions. Please refer to the attchment.

Reviewer 2 Report

This is a well-written review article, looking at different technologies for single-cell proteomic analysis. However, there are some concerns need to be addressed before publication,

(1) In the introduction, the authors should also discuss why it is necessary to measure the cellular proteins directly rather than using single-cell RNA-seq or single-cell ATAC-seq to infer the protein level in single cells?

(2) In 3.2 Droplet-Based Microfluidics, the authors should also mention CITE-seq (Stoeckius et al., Nature Methods 2017), which is an important technology for surface protein measurement at the single-cell level.

(3) In Conclusions and Outlook, the authors should also discuss single-cell multi-omics and spatial-omics, which are important future directions for single-cell proteomic analysis.

Author Response

(The authors gave the same response as above.)

Reviewer 3 Report

The review article of Liu and colleagues focuses on single-cell proteomic analysis.

The manuscript is interesting and well-organized although, in my opinion, it is "inhomogeneous". In fact, while the section that refers to Microfluidic approaches (paragraph 3 and sub-paragraphs) is well-discussed (a good number of applicatons being presented), that concerning the conventional approaches is very poor. In particular, sub-paragraphs 2,2; 2,3 and 2,4 (this latter in particular) mainly focus on the theoretical principles of the technicque although a few reference numbers are cited. However there is no discussion about the application of the cited authors.

I think the article should be reconsidered to make it more homogeneous.

A few language inaccuracies are listed below:

Line 98:..."fluorescence-labelled ..." should read "fluorescent-labelled.."

Line 101:..."applicable of.." ??

Line 112:..."due to that..."  ??

Line 164:".....utilizes capillary and electrophoretic techniques..." What does it mean ?

Line 208:"...microfabricated flow citometry.." should read "....flow citometer.."

Line 265 "Escherichia coli"  must be written in Italicc

Author Response

(The authors gave the same response as above.)

Round 2

Reviewer 1 Report

The manuscript has been improved well. This would be acceptable in this form. 

Reviewer 3 Report

The authors have introduced in the text  the modifications requested by the reviewer. In my opinion the quality of the article has improved and it is now worthy of publication on Cells.